# SARS-CoV-2 Accessory Protein ORF8 Targets the Dimeric IgA Receptor pIgR

**DOI:** 10.3390/v16071008

**Published:** 2024-06-22

**Authors:** Frederique Laprise, Ariana Arduini, Mathew Duguay, Qinghua Pan, Chen Liang

**Affiliations:** 1Lady Davis Institute, Jewish General Hospital, Montreal, QC H3T 1E2, Canada; frederique.laprise@mail.mcgill.ca (F.L.); ariana.arduini@mail.mcgill.ca (A.A.); mathew.duguay@gmail.com (M.D.); panqinghua07@gmail.com (Q.P.); 2Department of Microbiology and Immunology, McGill University, Montreal, QC H3A 2B4, Canada; 3Department of Medicine, McGill University, Montreal, QC H3G 2M1, Canada; 4Institut de Recherche Clinique de Montréal, Montreal, QC H2W 1R7, Canada

**Keywords:** SARS-CoV-2, ORF8, pIgR, IgA, IgM, mucosal immunity

## Abstract

SARS-CoV-2 is a highly pathogenic respiratory virus that successfully initiates and establishes its infection at the respiratory mucosa. However, little is known about how SARS-CoV-2 antagonizes the host’s mucosal immunity. Recent findings have shown a marked reduction in the expression of the polymeric Ig receptor (pIgR) in COVID-19 patients. This receptor maintains mucosal homeostasis by transporting the dimeric IgA (dIgA) and pentameric IgM (pIgM) across mucosal epithelial cells to neutralize the invading respiratory pathogens. By studying the interaction between pIgR and SARS-CoV-2 proteins, we discovered that the viral accessory protein Open Reading Frame 8 (ORF8) potently downregulates pIgR expression and that this downregulation activity of ORF8 correlates with its ability to interact with pIgR. Importantly, the ORF8-mediated downregulation of pIgR diminishes the binding of dIgA or pIgM, and the ORF8 proteins of the variants of concern of SARS-CoV-2 preserve the function of downregulating pIgR, indicating the importance of this conserved activity of ORF8 in SARS-CoV-2 pathogenesis. We further observed that the secreted ORF8 binds to cell surface pIgR, but that this interaction does not trigger the cellular internalization of ORF8, which requires the binding of dIgA to pIgR. These findings suggest the role of ORF8 in SARS-CoV-2 mucosal immune evasion.

## 1. Introduction

The Coronavirus Disease 2019 (COVID-19) pandemic, caused by SARS-CoV-2, has claimed over 7.03 million lives since its emergence in Wuhan, China, in 2019 [1,2]. Due to its high transmissibility, pathogenicity, and high mutational rate, SARS-CoV-2 continues to have a global impact. Moreover, the frequent emergence of variants of concern (VOCs) and variants of interest (VOIs) challenges the protective efficacy of the host immunity established either by COVID-19 vaccines or natural SARS-CoV-2 infections [3,4]. The clinical manifestation of COVID-19 is broad and depends on factors such as age, sex, and overall health condition [1,5,6]. The wide spectrum of symptoms has been shown to be attributed to differences in the proficiency of host immune responses.

Innate immune responses act as the first line of defence against viral infections [7]. One key player in this regard is interferon (IFN), which is induced upon the cellular detection of viral proteins and viral nucleic acids [7,8]. To successfully infect the host and spread, SARS-CoV-2 is able to use a group of its proteins to antagonize the IFN pathway, including the antagonization of viral RNA sensing, blockade of IFN signalling, shut-off of host translation, and obstruction of nuclear import and export [7]. Among these viral antagonization mechanisms is the viral accessory protein ORF8, which is unique as it is a secreted protein and bears an Ig-like domain, suggesting potentially novel viral mechanisms of systemic immunomodulation [9].

SARS-CoV-2 ORF8 has been of notable interest due to its large interactome network and its capacity to modulate host cellular pathways to promote immune evasion and viral replication [9,10]. Positioned among one of the most hypervariable regions in the SARS-CoV-2 genome following the spike protein, ORF8 serves as a strong recombination hotspot [11,12,13]. Despite its expression in some *Sarbecoviruses*, SARS-CoV-2 ORF8 only shares 55.4% nucleotide similarity with SARS-CoV ORF8 and 93% nucleotide similarity with bat-CoV RATG13 ORF8, the closest relative of SARS-CoV-2 identified to date [14]. Furthermore, SARS-CoV-2 ORF8 is structurally distinct from SARS-CoV ORF8 as the former forms a dimer that is held via an intermolecular disulfide bond (C20-C20), four salt bridges (D199-R115, R115-E92), and hydrogen bonds (F120-K53, K53-S24, Q18-L22, R52-I121) between the monomeric subunits [9,15,16,17,18,19]. Finally, ORF8 contains an Ig-like domain within its β-sandwich, which underlies its function as immune mimicry and its evasion of host immune pathways [12,15].

SARS-CoV-2 initiates its infection at the respiratory mucosa, which is protected by the host mucosal immunity [20]. An important mechanism of mucosal immunity is the production and secretion of dimeric IgA (dIgA) and pentameric IgM (pIgM), which bind and neutralize respiratory pathogens, including SARS-CoV-2 [21]. In response to infection, IgM is the first antibody produced; then, the production of IgA strengthens the neutralizing Ab responses during infection [22,23]. To be secreted, dIgA and pIgM must be transported from the lung subepithelial space to the mucosal lumen. This process, termed transcytosis, is mediated by the polymeric Ig receptor (pIgR) located at the basolateral surfaces of epithelial cells [24].

Human pIgR is a type I transmembrane protein that contains six extracellular domains, a transmembrane domain, and an intracellular domain [24]. The extracellular domain contains five Ig-like domains in tandem, which mediate the binding with dIgA and pIgM, while the sixth domain contains a proteolytic cleavage site that, upon cleavage by proteases, allows the release of sIgA after transcytosis [24,25]. When un-ligated, the ectodomain of pIgR, known as the secretory component (SC), adopts a closed conformation with primary interactions between domain (D) 1 and D5, D1 and D4, and finally D1 and D2 [26,27]. In its closed conformation, D1–D4–D5 forms a large interface with the respective complementarity determining regions (CDRs) facing outwards, suggesting that joining chain (JC) binding initially occurs with D1 and D5 [26,27]. This initial interaction promotes a conformational change within pIgR, separating D1 and D5 to allow D1 CDR1 to mediate the main interaction between SC and the JC of dIgA, while D3–D4–D5 becomes extended, leaving D4–D5 to have a minor interaction with both the Fc and JC of dIgA [26,27].

When SARS-CoV-2 infects the respiratory tracts and lungs, mucosal immunity is expected to suppress infection establishment through the transcytosis and secretion of dIgA and pIgM across the lung epithelial cells by pIgR, together with other immune mechanisms [28]. The loss of this biological function may render the airways more susceptible to SARS-CoV-2 infection, which is corroborated by the low pIgR levels in COVID-19 patients with severe disease [29,30]. To further understand the interplay between SARS-CoV-2 and pIgR, we have investigated which viral proteins may cause the downregulation of pIgR and discovered that ORF8 is capable of potently diminishing the expression of pIgR, thus potentially dampening the mucosal immunity by blocking the secretion of dIgA and pIgM; as a result, it promotes SARS-CoV-2’s infection of the respiratory epithelial cells.

## 2. Materials and Methods

### 2.1. Cell Culture

HEK293T cells (ATCC, cat. CRL-1573) were grown in Dulbecco’s Modified Eagle Medium (DMEM; ThermoFisher Scientific, Waltham, MA, USA) supplemented with 10% fetal bovine serum (FBS; ThermoFisher Scientific) and 1% penicillin and streptomycin (PS; ThermoFisher Scientific). Calu-3 lung epithelial cells (ATCC, cat. HTB-55) were grown in Eagle’s Minimum Essential Medium (EMEM; Wisent Bioproducts, St-Jean-Baptiste, Qc, Canada) supplemented with 20% FBS and 1% PS. Cells were passaged every second day or at 90% confluence using 0.05% Trypsin–EDTA (ThermoFisher Scientific, cat. 25300-054). Transfection was performed using polyethyleneimine (PEI; Sigma-Aldrich, St. Louis, MO, USA; cat. 913375), as previously described [31].

### 2.2. Plasmids

The lentiviral mammalian expression vector pLVX-E1alpha-IRES-Puro encoding WT SARS-CoV-2 ORF8 fused to a C-terminal Strep-II tag (Addgene, Watertown, MA, USA; cat. 141390) was used as an ORF8 expression plasmid [10]. Similarly, SARS-CoV-2 viral proteins encoded on the pLVX-E1alpha-IRES-Puro expression plasmid and fused to a C-terminal Strep-II tag were used (Addgene: E, cat. 141385; M, cat. 141386; N, cat. 141391; NSP1, cat. 141367; NSP2, 141368; NSP4, cat. 141369; NSP5, cat. 141370; NSP6, cat. 141372; NSP7, cat. 141373; NSP8, cat. 141374; NSP9, cat. 141375; NSP10, cat. 141376; NSP11, cat. 141377; NSP12, cat. 141378; NSP13, cat. 141379; NSP15, cat. 141381; ORF3a, 141383; ORF6, cat. 141387; ORF7a, cat. 141388; ORF9b, cat. 141392; ORF10, cat. 141394) [10]. In addition, we used a plasmid containing pIgR fused to the FLAG tag (GenScript, Rijswijk, The Netherlands; cat. OHu19522D), one having Flag-tagged FcRn (GenScript, cat. OHu24070D), and the empty vector pQCXIP (Addgene, cat. 631516). The lentiviral mammalian expression vector pLVX-E1alpha-IRES-Puro encoding SARS-CoV-2 ORF8-Strep-II was used as a backbone to generate ORF8 mutants, variants, and animal coronavirus ORF8, and the DNA fragments were synthesized by Genescript. We generated an empty vector plasmid from pLVX-E1alpha-IRES-Puro by excising ORF8-Strep-II using EcoRI (New England Biolabs, Ipswich, MA, USA; cat. R0101) and BamHI (New England Biolabs; cat. R0136). We transfected pLVX-EF1alpha-eGFP-2xStrep-IRES-Puro encoding eGFP (Addgene, cat. 141395), pMD2.G (expressing VSV-G; Addgene, cat. 12260), and psPAX2 (lentivirus packaging plasmid; Addgene, cat. 12260) to generate lentiviral vectors carrying ORF8 encoded by the pLVX-E1alpha-IRES-Puro expression vector, as previously described [31]. To generate pIgR domain deletions, a set of 6 primers was designed to amplify the extracellular-domain-deleted pIgR sequences (Appendix A). Amplification of the pIgR mutants was performed with polymerase chain reaction (PCR) using Accuprime Pfx DNA polymerase (Invitrogen, Waltham, MA, USA; cat. 12344024). The pIgR vector (pIgR-FLAG) was digested to remove the targeted region of pIgR using restriction enzymes HindIII (New England Biolabs; cat. R0104) and AfeI (New England Biolabs; cat. R0652). The extracted vector and amplified PCR products were ligated using T4 ligase (New England Biolabs; cat. M0202).

### 2.3. Western Blotting

Transfected cells were lysed in 50 mM Tris HCl, 150 mM NaCl, 1% NP-40, 0.5% sodium deoxycholate (SDS), 1 mM EDTA, 0.1% SDS, and 0.01% sodium azide (RIPA) supplemented with protease inhibitors (Sigma-Aldrich; cat. 118361700). Lysates were diluted in 4X Laemmli buffer and treated at 95 °C for 10 min. Protein samples were analyzed in a 12% sodium dodecyl sulfate (SDS; Bioshop, Burlington, ON, Canada; cat. SDS001.1) polyacrylamide gel by electrophoresis and transferred to a polyvinylidene difluoride (PVDF) membrane (Sigma-Aldrich; cat. 03010040001). The membranes were incubated with primary antibodies rat anti-Strep-II (1:5000; Abcam, Cambridge, UK; cat. Ab252885), mouse anti-FLAG (1:5000; Sigma-Aldrich; cat. F1804), and mouse anti-tubulin (1:5000; Santa Cruz Biotechnology, Dallas, TX, USA; cat. SC23948) or rabbit anti-SARS-CoV-2 spike protein (1:1000; Cell Signaling, Danvers, MA, USA; cat. 99423S) diluted in 2% bovine serum albumin (BSA; BioShop; cat. 9048-46-8) for 2 h at room temperature or with rabbit anti-LC3B (Abcam; cat. 192890) and rabbit anti-pIgR (1:1000; ThermoFisher Scientific; cat. PA5-3540) in 2% BSA overnight at 4 °C. Following this, the membranes were incubated with the secondary antibodies HRP-conjugated goat anti-rat (1:10,000; Invitrogen; cat. 31470), HRP-conjugated goat anti-rabbit (1:5000; SeraCare, Milford, MA, USA; cat. 5450-0010), or HRP-conjugated goat anti-mouse (1:5000; SeraCare; cat. 5450-0011) for 1 h at room temperature, and then exposed to enhanced chemiluminescence (ECL) reagents (PerkinElmer, Waltham, MA, USA; cat. NEL104001EA). The membranes were imaged by exposure to autoradiography films and quantified via the Fiji-ImageJ V2.14.0 software (NIH).

### 2.4. Co-Immunoprecipitation

Transfected cells were lysed in RIPA buffer supplemented with protease inhibitors and then incubated with MagStrep “type 3” XT beads (IBA-Lifesciences, Göttingen, Germany; cat. 2-4090-002) overnight at 4 °C or with anti-FLAG M2 Affinity Gel (Sigma-Aldrich; cat. A2220) for 2 h at 4 °C. Samples incubated with the Strep-II tag beads were placed in a magnetic Eppendorf tray and the supernatant was removed and conserved (flow-through), while samples incubated with FLAG beads were centrifuged at 6000 RPM for 5 min and the supernatants were removed and conserved (flow-through). The samples were washed and then precipitated under denaturing conditions by adding 1× Laemmli buffer and eluting the samples at 95 °C for 2 min. The supernatants were recovered and analyzed together with the whole cell lysates (WCL) via Western blot, as described above.

### 2.5. Antibody Conjugation

Recombinant SARS-CoV-2 ORF8-His (Invitrogen; cat. RP87666), IgA from human colostrum (Sigma-Aldrich; cat. I2636), and IgM from human serum (Sigma-Aldrich; cat. I8260) were conjugated using the Alexa Fluor 488 Microscale Protein Labelling Kit (Invitrogen; cat. A30006) and Alexa Fluor 647 Protein Labelling Kit (Invitrogen; A20173). Then, 1–1.5 mg/mL protein was conjugated by incubating it with 10% 1 M sodium bicarbonate alongside the reactive dye for 15 min or 1 h, as indicated in the company protocol. Conjugated proteins were purified using spin filters filled with the provided suspended gel resin and spun at 16,000× *g* for 1 min.

### 2.6. Flow Cytometry

Transfected cells were stained with cell viability stain Zombie Violet (1:2000) diluted in DBPS for 30 min at room temperature. Cells were washed three times with 3% BSA in DPBS and incubated with 10 μg/mL IgA, 3.3 μg/mL IgM, and/or 10 μg/mL ORF8 diluted in 3% BSA for 30 min at 4 °C, protected from light, as described in Appendix A. Cells were fixed with 4% paraformaldehyde (PFA; Bioshop; cat. PAR070.250) in DPBS for 15 min at room temperature. Cells were washed 1× BD permeabilizing wash buffer (BD Biosciences, Franklin Lakes, NJ, USA; cat. 51-2091KZ) in distilled H_2_O (dH_2_O). Cells were stained with either mouse Strep-II-FITC (1:1000; GenScript; cat. A01736-100) or anti-FLAG-647 (1:1000; Rockland Scientific, Victoria, BC, Canada; cat. 200-343-383) for 1 h at 4 °C. Cells were washed three times with 1× permeabilizing buffer and resuspended in a 200 μL mixture of 3% BSA and 1% PFA in DBPS and then stored at 4 °C until analysis on a BD FACSCanto Flow Cytometer.

### 2.7. Confocal Microscopy

Transfected cells were seeded on glass coverslips pre-treated with 10 μg/mL poly-D-lysine hydrobromide (Sigma-Aldrich; cat. P6407-5MG). Cells incubated with 10 μg/mL IgA-645 and/or 10 μg/mL ORF8-488 were diluted in 3% BSA for 30 min at 4 °C, followed by subsequent incubation at 37 °C for 15 min, protected from light, as described in Appendix A. Cells were fixed with 4% PFA for 10 min at room temperature. Cells were permeabilized with 0.1% Triton X-100 in DPBS for 10 min at room temperature and then blocked with 3% BSA for 1 h at room temperature. Cells were then stained with primary antibody rabbit anti-FLAG (1:200; Sigma-Aldrich; cat. 7425) diluted in 0.2% Triton X-100 and 1% BSA in DPBS for 2 h at room temperature. After washing, cells were stained with secondary antibody Alexa Fluor 568 goat anti-rabbit (1:500; Invitrogen; cat. A11016) diluted in 0.2% Triton X-100 and 1% BSA in DPBS for 1 h at room temperature. Cells were washed and finally stained with 1 μg/mL DAPI for 15 min at room temperature. Cells were imaged using a Zeiss (Oberkochen, Germany) LSM800 laser scanning confocal microscope.

### 2.8. Generating Calu-3 Cell Lines Stably Expressing ORF8 and Its Mutants

The reverse transduction of Calu-3 cells was performed to generate stable cell lines by preparing a 1:2 dilution of lentiviral vector aliquots in EMEM containing 10% FBS and 8 μg/mL hexadimethrine bromide (polybrene; Sigma-Aldrich; cat. H9268-10G). The 1:2 EMEM/lentiviral vector mix was added in a 1:1 volume ratio with Calu-3 cells and spinoculated in the centrifuge for 45 min at 1800 RPM. Cells were incubated for 48–72 h, until sufficient cell growth, and transduced cells were selected and maintained using EMEM containing 10% FBS, 1% PS, and 4 μg/mL puromycin dihydrochloride (Sigma-Aldrich; P8833-25MG). Cells were monitored and the medium was changed every two days until all control cells were killed, after which the confluent cells were expanded and maintained.

### 2.9. Data Quantification

All Western blotting data were quantified using Fiji–ImageJ V2.14.0. The relative intensity of the Western blots was measured and normalized to that of the control. The internalization of ORF8 to pIgR-transfected cells in confocal imaging was measured by Fiji–ImageJ V2.14.0. The PIgR and ORF8 channels were thresholds, and regions of interest were generated for cells expressing pIgR. The ratio of ORF8 binding and internalization to pIgR-positive and pIgR-negative cells was measured from these regions.

### 2.10. Statistical Analysis

All statistical analyses were performed using the GraphPad Prism V9 software. Correlation analysis was performed via correlation functions. Statistics were calculated using one-way ANOVA. Data shown are means of three independent experiments with standard deviations. Statistical significance is represented as follows: * *p* ≤ 0.05, ** *p* ≤ 0.01, *** *p* ≤ 0.001, **** *p* ≤ 0.0001.

## 3. Results

### 3.1. SARS-CoV-2 ORF8 Downregulates pIgR

In patients suffering from severe COVID-19 disease, the serum and urinary pIgR levels are markedly reduced [29,30]. This suggests SARS-CoV-2’s potential antagonization of pIgR-mediated mucosal immunity; however, the viral mechanisms behind this effect remain unknown. To investigate which viral proteins might be responsible for the decrease in pIgR, we co-transfected HEK293T cells with plasmids expressing pIgR and a panel of SARS-CoV-2 structural, non-structural, and accessory proteins, and we assessed the levels of pIgR expression by Western blots. Most of the SARS-CoV-2 proteins had minor upregulatory and downregulatory effects on pIgR expression, except for the spike glycoprotein, NSP1, and ORF8 (Figure 1A,B). NSP1 downregulated pIgR the strongest, due to its previously reported function as a potent inhibitor of translation; however, its intracellular location makes NSP1 unlikely to have systemic effects on pIgR outside infected cells (Figure 1B) [32,33]. Compared to the spike protein, which led to the mean downregulation of pIgR of 40%, ORF8 downregulated pIgR by 50% (Figure 1B). We further confirmed that ORF8 potently downregulated pIgR in a dose-dependent manner (Figure 1C,D). In contrast, we found that ORF8 did not downregulate the neonatal Fc receptor (FcRn), which belongs to the same Fc receptor family as pIgR, hence supporting the specificity of the ORF8-mediated downregulation of pIgR (Appendix A). The co-immunoprecipitation of ORF8 revealed an interaction between pIgR and ORF8, suggesting that ORF8 downregulates pIgR through interaction (Figure 1E). We next tested the lysosomal, proteasomal, and ER-associated degradation (ERAD) inhibitors on the ORF8-mediated downregulation of pIgR and did not observe a significant rescue effect (Appendix A) [34]. All four drugs are active, as shown by the increase in LC3B-II in the treated HEK293T cells (Appendix A), in agreement with the reported effect of these drugs in stimulating autophagy by different mechanisms [35,36,37,38]. We further investigated the possibility that ORF8 may increase the internalization of pIgR endocytosis, which then leads to the accumulation of intracellular pIgR and then promotes its downregulation. However, we did not observe the recovery of pIgR expression with the inhibition of endocytosis by dynasore, suggesting that ORF8 does not downregulate pIgR via increasing the internalization of the receptor (Appendix A).

### 3.2. ORF8 Proteins from SARS-CoV-2 VOCs and VOIs Preserve Their Downregulation of pIgR

ORF8 is hypervariable among SARS-CoV-2 variants and across beta-coronaviruses [12,13]. Hence, we first investigated whether the capacity for pIgR downregulation was conserved amongst ORF8 across different coronaviruses. Additionally, we sought to assess how the features of dimerization, multimerization, and glycosylation impact ORF8’s function, by generating a panel of mutants and variants derived from the WT SARS-CoV-2 ORF8 expression plasmid via mutagenesis (Figure 2A). Mutations C20A, R52A, K53A, and R115A are expected to disrupt ORF8 dimerization, Y73A disrupts multimerization, and N78A impairs glycosylation (Figure 2A). Moreover, we also cloned the Del119-120 ORF8 enriched in the Delta VOC, T11I in the Iota VOI, E92K in the gamma VOC, and V100L in the Epsilon VOI (Figure 2A). S24L and S84L were mutated based on their enrichment in SARS-CoV-2 ORF8 in the GSAID database (Figure 2A). Finally, we also tested SARS-CoV ORF8, along with ORF8 from SARS-CoV-like bat-CoV YNLF_31C and the SARS-CoV-2-like pangolin CoV, bat-CoV HKU3-7, and bat-CoV RatG13 (Figure 2A).

The co-expression of these different ORF8 proteins with pIgR revealed that RatG13, HKU3-7 (HKU3), Delta, Iota, Gamma, Epsilon, S84L, S24L, R52A, Y73A, and N78A downregulated pIgR to levels greater than or similar to the WT SARS-CoV-2 ORF8 (Figure 2B-C). Contrastingly, the ORF8 proteins from pangolin CoV, bat-CoV YNLF_31C, SARS-CoV, C20A, and R115A had a reduced capacity to downregulate pIgR (Figure 2B,C). These results suggest a potential role of ORF8 dimerization in the downregulation of pIgR, since two out of the four dimerization mutants (C20A and R115A) and the monomeric ORF8 proteins of SARS-CoV and bat-CoV YNLF_31C exhibited the weaker downregulation of pIgR (Figure 2B,C). Similarly, the stable expression of these ORF8 mutants and variants in Calu-3 cells led to the downregulation of endogenous pIgR (Appendix A). We next examined the interaction of these ORF8 mutants and variants with pIgR by performing co-immunoprecipitation and observed that the ORF8 proteins that displayed weaker downregulation of pIgR, notably pangolin CoV, bat-CoV YNLF_31C, SARS-CoV, and C20A, showed decreased binding to pIgR relative to the WT SARS-CoV-2 ORF8 (Figure 2D,E). In contrast, R115A ORF8 did not show a reduced capacity to interact with pIgR, consistent with its capacity to effectively downregulate pIgR (Figure 2D,E). We further performed a correlation analysis between the relative ORF8 binding to pIgR and the relative pIgR downregulation. The data confirmed that the ORF8 mutants that bound pIgR more strongly, such as SARS-CoV-2 and R115A, caused the greater downregulation of pIgR, while mutants that bound pIgR less strongly, such as C20A, pangolin, and YNLF ORF8, poorly downregulated pIgR (Figure 2A–F). Overall, these results demonstrate that the downregulation of pIgR is a conserved function of ORF8 from different coronaviruses and SARS-CoV-2 variants and that the level of pIgR downregulation correlates with the degree of interaction with ORF8.

### 3.3. SARS-CoV-2 ORF8 Downregulation of pIgR Attenuates Cellular Binding of dIgA and pIgM

To assess the functional impact of ORF8’s downregulation of pIgR, we investigated whether ORF8 expression inhibited the binding of dIgA or pIgM to the cell surface pIgR. Hence, we incubated HEK293T cells co-expressing pIgR and ORF8 with dIgA or pIgM and measured their binding by flow cytometry. As expected, we found that intracellular ORF8 decreased the binding of both total dIgA and pIgM to pIgR-expressing cells (Figure 3). Specifically, ORF8 reduced the mean dIgA binding to pIgR-expressing cells from 59.6% in control cells to 36.9% in ORF8-expressing cells (Figure 3A,B). As for pIgM, the mean binding to pIgR-expressing cells dropped from 33.9% in control cells to 25.1% as a result of ORF8 expression (Figure 3C,D). These results support the capacity of intracellular ORF8 to diminish the binding of dIgA or pIgM to pIgR.

### 3.4. ORF8 Downregulates pIgR Mutants Deleted of the Extracellular Domains

We next investigated which domains of pIgR mediated downregulation by ORF8 by generating a panel of pIgR mutants with the deletion of the extracellular domains (Figure 4A). Since ORF8 is a luminal protein and secreted, it more likely interacts with pIgR’s extracellular domains [39,40,41]. We therefore removed these extracellular domains in a sequential manner and generated deletions of domain 1 (ΔD1), domains 1 through 2 (ΔD1–2), domains 1 through 3 (ΔD1–3), domains 1 through 4 (ΔD1–4), and domains 1 through 5 (ΔD1–5) (Figure 4A). When these mutants were expressed in HEK293T cells, ΔD1, ΔD1–3, and ΔD1–4 were very poorly expressed ΔD1–2, and D1–5 showed sufficiently high levels of expression for the test of their sensitivity to downregulation by ORF8. Interestingly, both ΔD1–2 and ΔD1–5 were downregulated by ORF8 similarly to the WT pIgR (Figure 4B,C). Consistent with its weak downregulation of the WT pIgR, the C20A ORF8 mutant did not markedly affect the expression levels of ΔD1–2 and ΔD1–5 (Figure 4D,E). Therefore, the folded extracellular domains of pIgR are dispensable for ORF8-mediated downregulation, and C20-mediated ORF8 dimerization is critical in downregulating pIgR.

### 3.5. Secreted ORF8 Binds to Cell Surface pIgR

Since ORF8 is secreted, we thus investigated whether it could interact with pIgR at the surfaces of un-infected cells and interfere with pIgR function. To this end, we incubated pIgR-expressing HEK293T cells with Alexa-488-conjugated recombinant ORF8 at 4 °C to allow ORF8’s binding to the cell surface proteins, and we then examined the treated cells by flow cytometry (Appendix A). The strong binding of ORF8 to pIgR-expressing cells, but not to the control cells, was recorded (Figure 5A). We next examined whether IgA competed with ORF8 for binding to pIgR. Increasing amounts of un-conjugated IgA were co-incubated with ORF8 and pIgR-expressing cells, and ORF8’s binding decreased from 29% to 18% when 10 ug/mL IgA was added (Appendix A), suggesting that IgA is able to partially inhibit the binding of ORF8 to cell surface pIgR. To determine which extracellular domain(s) of pIgR is bound by ORF8, we incubated Alexa-488-conjugated ORF8 with HEK293T cells expressing the pIgR mutants described above. Consistent with the results of the Western blots, the ΔD1, ΔD1–3, and ΔD1–4 mutants were less well expressed compared to ΔD1–2 and ΔD1–5 (Appendix A). ORF8 did not show detectable binding to any of these pIgR mutants (Figure 5), suggesting that either D1 serves as the binding site or a tertiary structure formed by multiple domains is required for ORF8 binding. In fact, in silico and in vitro analyses have shown that the main interactions of the pIgR domains are mediated by the binding of D1 to D2, D4, and D5, and the binding of D2 and D3, as well as D4 and D5 [27].

### 3.6. ORF8 Is Internalized Together with the pIgR–IgA Complex

ORF8’s binding to cell surface pIgR may trigger the internalization and loss of pIgR from the cell surface. To test this possibility, we performed confocal imaging to monitor this process. We first treated pIgR-expressing HEK293T cells with Alexa-647-conjugated dIgA and Alexa-488-conjugated ORF8, either individually or together at 4 °C (Appendix A). The results showed the cell-peripheral localization of pIgR, while ORF8 and dIgA individually co-localized with cell surface pIgR (Figure 6A). When both ORF8 and dIgA were present, they were found to be either co-localized or separate from one another, suggesting that they can interact with the same pIgR cluster (Figure 6A).

We then washed off the unbound ORF8 or dIgA and switched cells to 37 °C to allow endocytosis to occur (Appendix A). After 15 min incubation, ORF8 or dIgA was individually localized and agglomerated close to the plasma membrane (Figure 6B). Similarly, the co-incubation of ORF8 and dIgA together led to large punctates near the plasma membrane, likely indicating the localization of both proteins to the same pIgR-rich regions in preparation for internalization (Figure 6B). This is in agreement with early studies reporting that dIgA binding leads to pIgR’s homotypic dimerization and crosslinking, which can trigger the internalization of the dIgA–pIgR complex [42]. Continued incubation at 37 °C for 30 min did not permit the internalization of ORF8 but did lead to the observation of small intracellular dIgA punctates overlapping with the pIgR signals, suggesting the successful endocytosis of the dIgA–pIgR complex but not ORF8 (Figure 6C). Interestingly, when ORF8 was co-incubated with dIgA, large intracellular punctates containing ORF8, dIgA, and pIgR were recorded (Figure 6C). These intracellular IgA/ORF8 signals were not detected when dynasore was used to block endocytosis (Figure 6). Furthermore, we confirmed that ORF8 was three to six times more likely to bind pIgR-expressing cells than pIgR-naïve cells, especially in the presence of dIgA (Figure 6D). These results suggest that ORF8 can bind to cell surface pIgR, but this binding itself does not trigger pIgR’s internalization; however, ORF8 can enter cells together with the dIgA–pIgR complex.

## 4. Discussion

The mucosa is the main site of infection by many pathogens. To successfully invade the mucosa and establish infection, pathogens have evolved various strategies to antagonize and even hijack the host mucosa immune mechanisms. In this study, we report that SARS-CoV-2 uses its accessory protein, ORF8, to downregulate a key factor in mucosal immunity, namely pIgR, and that pIgR’s downregulation is expected to decrease the secretion of mucosal IgA and dampen mucosal immunity. Indeed, our study showed that the SARS-CoV-2 ORF8-mediated downregulation of pIgR results in decreased dIgA binding to cells. Interestingly, we observed a trend whereby dimerization-deficient ORF8 proteins (SARS-CoV, YNLF, C20A, and R115A) were less efficient in downregulating pIgR in correlation with their poor binding to pIgR. Since dimerization is a unique feature that SARS-CoV-2 ORF8 has acquired but that is absent in the ORF8 of SARS-CoV and other coronaviruses, it is speculated that SARS-CoV-2 ORF8 may be more effective in dampening pIgR-mediated mucosal immunity and thus better assist viral infection and spread.

ORF8 has been reported to downregulate other important immune molecules, including MHC-I [34,43]. Although ORF8 interacts with both pIgR and MHC-I in order to downregulate these cellular proteins, the downregulation of MHC-I requires the functional autophagy/lysosomal pathways [34], whereas the inhibition of lysosomes and autophagosomes did not affect the ORF8-mediated downregulation of pIgR. In addition to ORF8, SARS-CoV-2 employs other proteins, such as NSP1 and the spike, to decrease pIgR expression (Figure 1A,B), a phenomenon that was also reported for the downregulation of MHC-I by not only ORF8 but also by viral proteins ORF3a and ORF7a [43,44,45]. This highlights the importance of downregulating these immune molecules in assisting the viral evasion of host immunity and promoting viral infection and pathogenesis.

One unique characteristic of ORF8 is that it is a secretory protein that has been detected in the blood of SARS-CoV-2-affected individuals and thus has the capacity to exert systemic effects on cells that are distal to the primary site of infection [46]. Our data showed that the secreted ORF8 can bind to cell surface pIgR. Interestingly, this binding itself does not appear to trigger ORF8’s internalization, likely because ORF8’s binding is not sufficient to cause pIgR clustering. However, when dIgA is present and binds to pIgR, ORF8 can attach to the dIgA–pIgR complex and enter cells. This suggests a mechanism by which ORF8, after being secreted by SARS-CoV-2-infected cells, may enter naïve, uninfected cells to exert its immunomodulatory functions and render these cells more amenable to infection. Overall, our results suggest that intracellular ORF8 contributes to the viral downregulation of pIgR and that the secreted ORF8 can bind to cell surface pIgR and enter cells together with bound dIgA (Figure 7). The functions of the internalized ORF8 await further investigation.

SARS-CoV-2 is not the only virus that can downregulate pIgR. Other mucosal viral pathogens, including SIV and S/HIV, have been reported to decrease pIgR levels [47,48]. Some pathogens have even evolved to hijack pIgR to promote viral infection. For example, Epstein–Barr virus (EBV) and *S. pneumoniae* use the dIgA–pIgR complex to assist entry into target cells [49,50,51]. It is unknown to what extent SARS-CoV-2 may assimilate pIgR to promote its infection—for example, whether the viral spike protein shows any affinity for pIgR. Regardless, our study does show that ORF8 exploits the dIgA–pIgR complex to enter cells.

Our study has several limitations. ORF8 may not be the only viral protein that is able to downregulate pIgR, and its relative contribution to this important viral function remains to be determined using ORF8-deleted SARS-CoV-2. In this context, it is known that SARS-CoV-2 employs host shut-off mechanisms to promote efficient viral replication and production [52]. As such, the downregulation of pIgR may be one of the many outcomes of the viral antagonization of host protein synthesis, as observed with NSP1, which contributes to the dampening of host antiviral defence, including mucosal immunity [32]. Similarly, whether the ORF8-mediated downregulation of pIgR is in part attributed to its the induction of the unfolded protein response or its wide interactome warrants further investigation [10,40,53].

Furthermore, we observed the ORF8-mediated downregulation of pIgR; however, the extent to which this can impair the secretion of dIgA needs to be further investigated. This latter task will require the development of a sensitive dIgA transcytosis assay using polarized epithelial cells, which has been a challenge in the field. Lastly, it will be interesting to investigate the functions of the internalized ORF8 protein, particularly whether it can exert the reported immunomodulation functions, such as impairing IFN pathways and acting as histone mimics. Despite these limitations, our work provides insights into SARS-CoV-2’s mechanisms to evade host mucosal immunity and expands our understanding of the multifaceted immunomodulatory functions of ORF8, from downregulating MHC-I to pIgR.

## Figures and Tables

**Figure 1 viruses-16-01008-f001:**
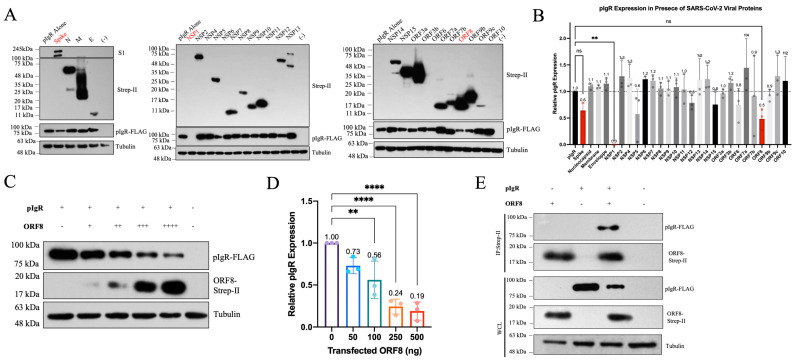
SARS-CoV-2 ORF8 downregulates pIgR in a dose-dependent manner. (**A**,**B**) HEK293T cells co-transfected with 500 ng pIgR plasmid DNA; 250 ng plasmid DNA expressing SARS-CoV-2 structural, non-structural (nsp), or accessory proteins; and 250 ng QCXIP empty vector in 6-well plates. (**C**,**D**) HEK293T cells co-transfected with 500 ng pIgR plasmid DNA and a titration of SARS-CoV-2 ORF8-Strep-II plasmid DNA (0 ng (-), 50 ng (+), 100 ng (++), 250 ng (+++), 500 ng (++++)) and QCXIP DNA. (**E**) HEK293T cells co-transfected with 2500 ng pIgR, 1250 ng ORF8-Strep-II, and 1250 ng QCXIP plasmid DNA in a 10 cm dish. Whole cell lysates were harvested and analyzed for pIgR-FLAG (anti-FLAG), SARS-CoV-2 proteins, and ORF8-Strep-II (anti-Strep-II), spike (anti-spike), and tubulin (anti-tubulin) by Western blots. Protein expression was quantified using Fiji and analyzed with Prism V9 (mean with SD; statistical significance measured via one-way ANOVA; ns (not significant) *p* ≥ 0.05, ** *p* ≤ 0.01, **** *p* ≤ 0.0001).

**Figure 2 viruses-16-01008-f002:**
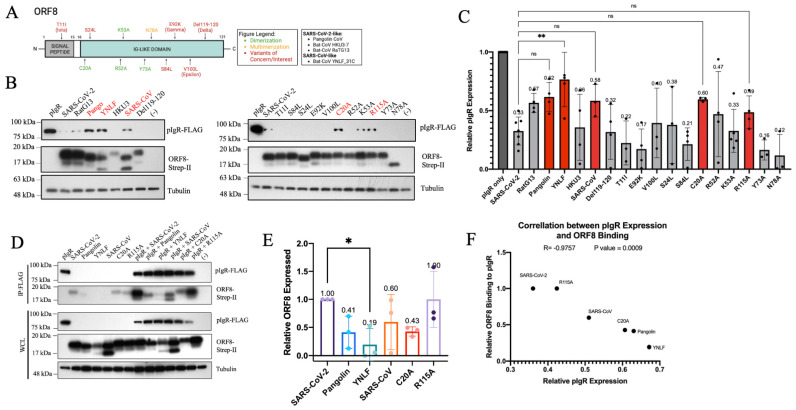
ORF8 from SARS-CoV-2 mutants, variants, and animal CoVs interact differently with pIgR. (**A**) Schematic of ORF8 mutagenesis. (**B**,**C**) HEK293T co-transfected with 500 ng pIgR; 250 ng QCXIP; and 250 ng mutant, variant, and animal CoV ORF8 proteins. (**D**,**E**) HEK293T co-transfected with 2500 ng pIgR; 1250 ng QCXIP; and 1250 ng pangolin, YNLF, SARS-CoV, C20A, and R115A ORF8 proteins. (**F**) Correlation analysis performed on PrismV9. (**B**–**E**) Protein expression was quantified using Fiji and analyzed with Prism V9 (mean with SD; statistical significance measured via one-way ANOVA; ns (not significant) *p* ≥ 0.05, * *p* ≤ 0.05, ** *p* ≤ 0.01).

**Figure 3 viruses-16-01008-f003:**
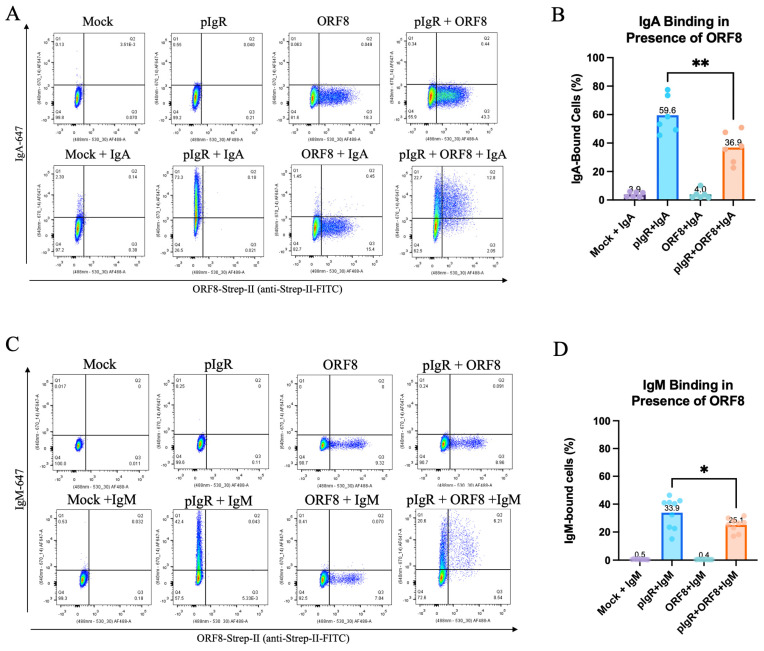
Intracellular ORF8 restricts dIgA and pIgM binding to pIgR. (**A**–**D**) HEK293T cells co-transfected with 250 ng pIgR, 250 ng QCXIP, and 500 ng SARS-CoV-2 ORF8 expression plasmids. Cells incubated with 10 μg/mL IgA-647 (**A**,**B**) or 3.3 μg/mL IgA-647 (**C**,**D**) for 30 min on ice and stained for ORF8-Strep-II (anti-Strep-II-FITC). IgA and IgM binding was assessed by flow cytometry. Data were analyzed with Prism V9 (mean with SD; statistical significance measured via one-way ANOVA; * *p* ≤ 0.05, ** *p* ≤ 0.01).

**Figure 4 viruses-16-01008-f004:**
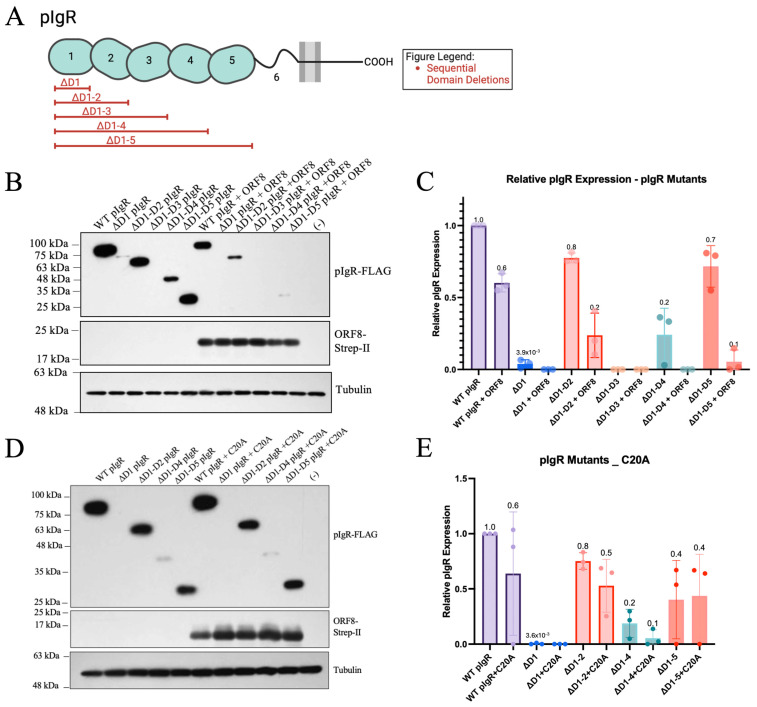
Intracellular ORF8 downregulates pIgR mutants. (**A**) Schematic of pIgR mutants. (**B**,**C**) HEK293T cells co-transfected with 500 ng pIgR or pIgR mutants, 250 ng ORF8, and 250 ng QCXIP. (**D**,**E**) HEK293T co-transfected with 500 ng pIgR or pIgR mutants, 250 ng QCXIP, and 250 ng C20A ORF8. (**B**–**E**) Whole cell lysates were harvested and analyzed for pIgR-FLAG (anti-FLAG), ORF8-Strep-II (anti-Strep-II), and tubulin (anti-tubulin) by Western blots. Protein expression was quantified using Fiji and analyzed with Prism V9 (mean with SD; statistical significance measured via one-way ANOVA.

**Figure 5 viruses-16-01008-f005:**
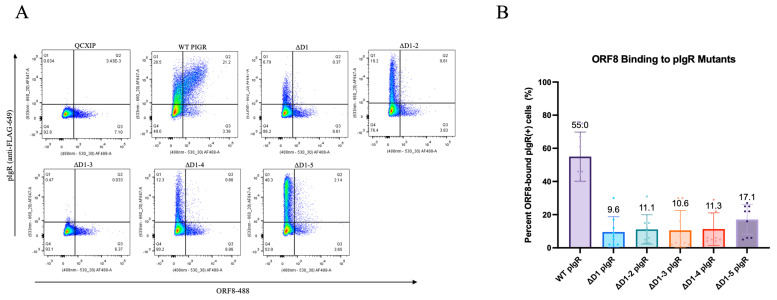
Secreted ORF8 interacts with D1 of pIgR. (**A**,**B**) HEK293T cells were transfected with 250 ng pIgR or pIgR mutants and treated with 10 μg/mL ORF8-488 for 30 min on ice. Cells were stained with anti-FLAG (pIgR) and analyzed by flow cytometry. Data were analyzed by PrismV9 (mean with SD; statistical significance was measured via one-way ANOVA).

**Figure 6 viruses-16-01008-f006:**
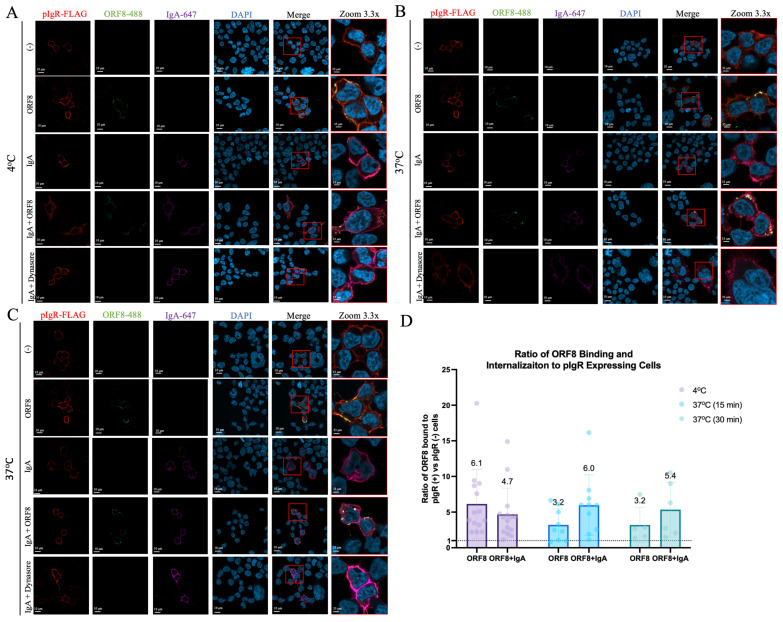
Secreted ORF8 enters cells together with the pIgR–IgA complex. (**A**–**D**) HEK293T cells were co-transfected with 250 ng pIgR and 250 ng QCXIP expression plasmids and then treated with 10 μg/ml IgA-647 (purple) and/or 10 μg/ml ORF8-488 (green) for 30 min on ice (**A**). Cells were washed and incubated at 37 °C for 15 (**B**) and 30 min (**C**). Cells were stained for pIgR-FLAG (anti-FLAG; red) and DAPI (blue). Cellular localization was visualized by confocal microscopy. (**D**) A macro program was developed to analyze and quantify the ratios of ORF8 bound and internalized to pIgR-expressing vs. pIgR-naïve cells. Data were analyzed with Prism V9 (mean with SD; statistical significance measured via one-way ANOVA.

**Figure 7 viruses-16-01008-f007:**
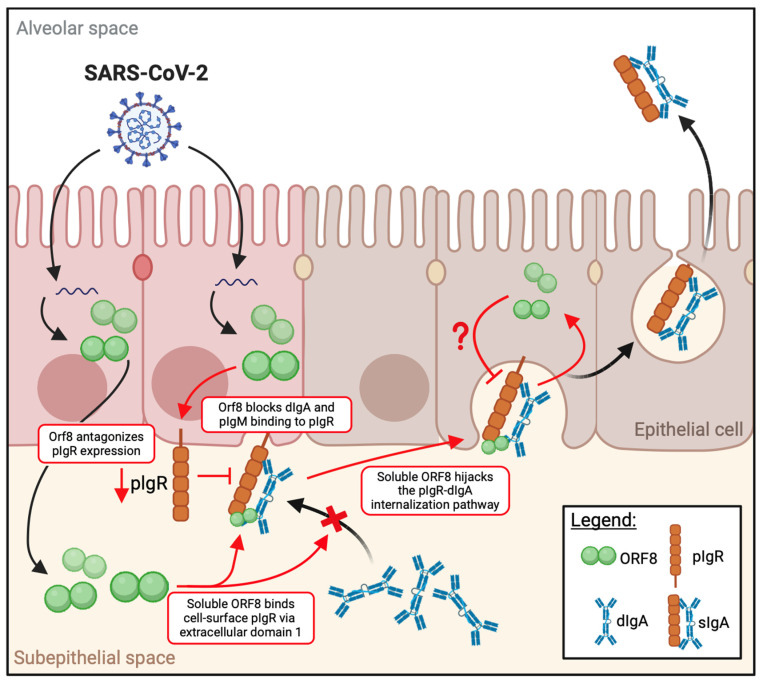
SARS-CoV-2 ORF8 modulates pIgR expression and highjacks the dIgA–pIgR pathway for internalization. Graphical abstract demonstrating the main pathways of pIgR modulation by SARS-COV-2 ORF8. Within cells, ORF8 downregulates pIgR expression, leading to decreased cell surface dIgA binding. In contrast, soluble ORF8 interacts with and highjacks the dIgA–pIgR complex to enter cells (as indicated by the red arrows and “× ”). Whether the immunomodulation of pIgR by ORF8 leads to an antagonized sIgA response remains to be investigated (as indicated with “?”). Created with biorender.com (accessed on 16 April 2024).

## Data Availability

All data in this manuscript are publicly available.

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
