# Peer review of "SARS-CoV-2 Accessory Protein ORF8 Targets the Dimeric IgA Receptor pIgR"

_viruses, 2024, doi:10.3390/v16071008_

Round 1

Reviewer 1 Report

Comments and Suggestions for Authors

To the Authors,
The manuscript is really well written, the topic is intriguing, and the results explained clearly.
The Introduction and Discussion sections are fluent and captivating (which is not easy after almost five years of SARS-CoV-2 related papers). I appreciate that the statistic section (which is often neglected) is precisely reported.
I think that this manuscript is of interest for the scientific community as it shears light upon an aspect which has been less investigated through these years of pandemic.
However, in addition to some minor things that could be easily fixed (listed below), my major concern regards the use of transient co-transfections as principal method to assess the effect of SARS-CoV-2 proteins on pIgR expression and to evaluate the efficacy of ORF8 mutants.
I recommend the Authors to address my points in order to clarify my doubts and make the manuscript suitable for publication.

Major
It is difficult to assess the effect of SARS-CoV-2 proteins on pIgR expression or the efficacy of ORF8 mutants (or ORF8 derivedfrom different viral strains) through co-transfection experiments, as this approach is not internally controlled. Each co-transfection is an independent experiment and has its own yield and efficiency which can be a lot different between experiments. The pIgR expression could be different between sample “a” and “b” due to transfection itself and not due to viral protein expression. Furthermore, the expression of viral protein per se impacts protein expression machineries in general, so the effect on pIgR expression could be unspecific. In our hand co-expression of SARS-CoV-2 proteins together with “other” proteins, dampens the expression of many of the “others” no matter the protein being transfected.
The experiment shown in Figure S2, indeed, strengthen my doubts. In Calu-3 cells stably expressing ORF8 (thus, avoiding co-transfection), the reduction of pIgR levels is way different than that shown in Figure 1 and 2. It is true that HEK293T and Calu-3 are two different cell line, but since Calu-3 are widely used (together with Vero-E6 and A549) as the main cell line for SARS-CoV-2 experiments, it would be worth to use Calu-3 to perform control experiment that I suggest the Authors to do.

Hence, since the starting point of this work is that ORF8 expression reduces pIgR levels and the Authors then evaluate the effect of ORF8 mutants in the same way (transient co-transfections), they should provide the proper controls for this approach.
The Authors should perform an experiment in which pIgR expression would not be affected by transfection per se, for example: i) use a cell line which expresses endogenous pIgR; ii) use HEK293T stably expressing tagged-pIgR. Otherwise, the Authors could co-transfect a panel of structural and non-structural viral proteins together with a reporter protein in Calu-3 cells, to evaluate if its expression is altered or not. The Authors could show the results of one of the approaches I suggested in a supplementary figure.
I am afraid that if not addressed, this approach weakens, in my opinion, the data about binding and internalization shown in the next figures.

- Since is not reported in the Methods section (please report it), I assume that all SARS-CoV-2 proteins are expressed as C-terminal strep-tagged, but for some of them it has been reported that C-terminal tagging alters the physiological behavior. This is the case of NSP6 for instance. Have the Authors evaluated pIgR levels also upon the expression of N-ter tagged viral proteins?
Since it has been demonstrated that NSP6
antagonizes the interferon type I (IFN-I) responses, and NLRP3 inflammasome, it would be of interest evaluate if it plays a role in the pIgR expression.

- Why NSP3 has not been transfected?

- The authors must show that the drugs used in the experiments reported in Figure S1a and S1b are effectively working, through western blot analysis of known lysosomal, proteasomal and ERAD markers.

Minor
- The subpanels notation in Figures is in uppercase, while in the caption is in lowercase.
- In figure 1 subpanel “e” legend is reported as “f” and does not indicate that a co-IP is shown.
- In Figure 1A right panel overlaps in part the middle one.
- In Figure 1B non-significant p-values are reported as numbers, while in Figure 2C are reported as “ns”, please be consistent among figures.
- The anti-Strep II blot in the middle Figure 1A is not perfectly aligned to that of tubulin and pIgR, making difficult to see which lane refers to the labels. It seems, for instance, that the NSP6 lane is empty, while there is a strong reduction of pIgR signal.

Author Response

Reviewer #1

The manuscript is really well written, the topic is intriguing, and the results explained clearly.
The Introduction and Discussion sections are fluent and captivating (which is not easy after almost five years of SARS-CoV-2 related papers). I appreciate that the statistic section (which is often neglected) is precisely reported.
I think that this manuscript is of interest for the scientific community as it shears light upon an aspect which has been less investigated through these years of pandemic.
However, in addition to some minor things that could be easily fixed (listed below), my major concern regards the use of transient co-transfections as principal method to assess the effect of SARS-CoV-2 proteins on pIgR expression and to evaluate the efficacy of ORF8 mutants.
I recommend the Authors to address my points in order to clarify my doubts and make the manuscript suitable for publication.

Response: We would like to express our gratitude to reviewer #1 for the detailed comments on our submission, they have been invaluable in helping to refine and improve the quality of our work. In the following sections, we provided our detailed responses.

Major
-It is difficult to assess the effect of SARS-CoV-2 proteins on pIgR expression or the efficacy of ORF8 mutants (or ORF8 derived from different viral strains) through co-transfection experiments, as this approach is not internally controlled. Each co-transfection is an independent experiment and has its own yield and efficiency which can be a lot different between experiments. The pIgR expression could be different between sample “a” and “b” due to transfection itself and not due to viral protein expression. Furthermore, the expression of viral protein per se impacts protein expression machineries in general, so the effect on pIgR expression could be unspecific. In our hand co-expression of SARS-CoV-2 proteins together with “other” proteins, dampens the expression of many of the “others” no matter the protein being transfected.
The experiment shown in Figure S2, indeed, strengthen my doubts. In Calu-3 cells stably expressing ORF8 (thus, avoiding co-transfection), the reduction of pIgR levels is way different than that shown in Figure 1 and 2. It is true that HEK293T and Calu-3 are two different cell line, but since Calu-3 are widely used (together with Vero-E6 and A549) as the main cell line for SARS-CoV-2 experiments, it would be worth to use Calu-3 to perform control experiment that I suggest the Authors to do.
Hence, since the starting point of this work is that ORF8 expression reduces pIgR levels and the Authors then evaluate the effect of ORF8 mutants in the same way (transient co-transfections), they should provide the proper controls for this approach.
The Authors should perform an experiment in which pIgR expression would not be affected by transfection per se, for example: i) use a cell line which expresses endogenous pIgR; ii) use HEK293T stably expressing tagged-pIgR. Otherwise, the Authors could co-transfect a panel of structural and non-structural viral proteins together with a reporter protein in Calu-3 cells, to evaluate if its expression is altered or not. The Authors could show the results of one of the approaches I suggested in a supplementary figure.
I am afraid that if not addressed, this approach weakens, in my opinion, the data about binding and internalization shown in the next figures.

Response: We agree with the concerns of this reviewer regarding the proper controls to assure that the downregulation of pIgR by SARS-CoV-2 ORF8 is reproducible and specific. To control for the possible variation of transfection efficiency between individual transfections, all ORF8 co-transfection experiments have been performed at least three times, the results of multiple transfections were analyzed and presented in the bar graphs (Figure 1B, 1D). And downregulation of pIgR by ORF8 was consistently observed across the different transfections. In addition, one control experiment that we did to further address this concern is to measure the effect of ORF8 on Fc neonatal receptor (FcRn) which belongs to the same family of Fc receptors as pIgR. In this assay, ORF8 did not affect the expression of FcRn (Figure S2A), which supports the specificity of ORF8-mediated downregulation of pIgR. The data are described in lines 246 to 248, “In contrast, we found that ORF8 did not downregulate the neonatal Fc receptor (FcRn), which belongs to the same Fc receptor family as pIgR, hence supporting the specificity of ORF8-mediated downregulation of pIgR (Figure S2A).”

We also acknowledge that SARS-CoV-2 employs the host shut-off mechanism to limit the synthesis of host proteins in order to promote viral replication and production as well as to dampen host antiviral responses [ref 52]. In this context, pIgR cannot be the only cellular protein that is downregulated by SARS-CoV-2 proteins including NSP1 and ORF8. This is further discussed in lines 473-480: “In this context, it is known that SARS-CoV-2 employs host shut-off mechanisms to promote efficient viral replication and production [52]. As such, downregulation of pIgR may be one of the many outcomes from the viral antagonization of host protein synthesis, as observed with NSP1, which contributes to the dampening of host antiviral defense including mucosal immunity [32]. Similarly, whether ORF8-mediated downregulation of pIgR is in part attributed to its the induction of the unfolded protein response, or its wide interactome, warrants further investigation [10,40,53].”

In regard to the pIgR downregulation data in Calu-3 cells, this was not acquired by co-transfection, mainly because high transfection efficiency of Calu-3 is not easy to achieve. Instead, we generated ORF8-expressing Calu-3 cell lines. Even so, only 40-50% of the total Calu-3 cell populations stably expressed ORF8 (Figure S3A), and therefore 50-60% of Calu-3 cells per sample were unaffected by ORF8, which compromises the reliable evaluation of ORF8-mediated downregulation of endogenous ORF8 and partly explains the relatively weak decrease of pIgR as shown in Figure S3B and S3C.

- Since is not reported in the Methods section (please report it), I assume that all SARS-CoV-2 proteins are expressed as C-terminal strep-tagged, but for some of them it has been reported that C-terminal tagging alters the physiological behavior. This is the case of NSP6 for instance. Have the Authors evaluated pIgR levels also upon the expression of N-ter tagged viral proteins?
Since it has been demonstrated that NSP6 antagonizes the interferon type I (IFN-I) responses, and NLRP3 inflammasome, it would be of interest evaluate if it plays a role in the pIgR expression.

- Why NSP3 has not been transfected?

Response: We obtained all plasmids expressing SARS-CoV-2 proteins from Addgene that were deposited by Gordon DE et al [10], the NSP3 DNA construct was not included in this panel of plasmids, we therefore did not test NSP3 against pIgR. We did co-transfect NSP6 with pIgR and did not observe any marked effect on pIgR expression (Figure 1A, 1B ).

All these plasmids have the Strep-II tag attached to the C-terminus of each SARS-CoV-2 protein, which is now explicitly described in Materials and Methods, lines 108-114, “Similarly, SARS-CoV-2 viral proteins encoded on pLVX-E1alpha-IRES-Puro expression plasmid and fused to a C-terminal Strep-II tag were used  (Addgene: E, cat. 141385; M, cat. 141386; N, cat. 141391; NSP1, cat. 141367; NSP2, 141368; NSP4, cat. 141369; NSP5, cat. 141370; NSP6, cat. 141372; NSP7, cat. 141373; NSP8, cat. 141374; NSP9, cat. 141375; NSP10, cat. 141376; NSP11, cat. 141377; NSP12, cat. 141378; NSP13, cat. 141379; NSP15, cat. 141381; ORF3a, 141383; ORF6, cat. 141387; ORF7a, cat. 141388; ORF9b, cat. 141392; ORF10, cat. 141394).”

We agree that the position of the tag may have an impact of the function of the protein such as ORF8. Since ORF8 has the N-terminal signal peptide, we are limited to attached the tag to its N-terminus which may affect ORF8 secretion. In addition, the C-terminally tagged ORF8 has been successfully used in different studies [10,53] .

- The authors must show that the drugs used in the experiments reported in Figure S1a and S1b are effectively working, through western blot analysis of known lysosomal, proteasomal and ERAD markers.

Response: We have now provided data to show that the compounds used in Figure S2B (previous S1b) are active. Since all these four compounds has been reported in the literature to increase the level of LC3B-II [35-38], a key autophagic factor, we thus treated HEK293T cells with each of the four drugs and measured LC3B-II by Western blotting. The results are presented in Figure S2C, described in line 253 to 255, “All four drugs are active as shown by the increase of LC3B-II in the treated HEK293T cells (Figure S2C), in agreement with the reported effect of these drugs on stimulating autophagy by different mechanisms [35-38].”

Minor
- The subpanels notation in Figures is in uppercase, while in the caption is in lowercase.

Response: Subpanels notation in the caption is now in uppercase in all figures.

- In figure 1 subpanel “e” legend is reported as “f” and does not indicate that a co-IP is shown.

Response: Subpanel “f” of Figure 1 is now changed to “e” to match the figure.

- In Figure 1A right panel overlaps in part the middle one.

Response: The overlapping has now been corrected.

- In Figure 1B non-significant p-values are reported as numbers, while in Figure 2C are reported as “ns”, please be consistent among figures.

Response: Significance in Figure 1B has now been changed from being reported in numbers to Asterix and “ns”.

- The anti-Strep II blot in the middle Figure 1A is not perfectly aligned to that of tubulin and pIgR, making difficult to see which lane refers to the labels. It seems, for instance, that the NSP6 lane is empty, while there is a strong reduction of pIgR signal.

Response: We have now re-aligned the SARS-CoV-2 protein blots with both tubulin and pIgR in Figure 1A, to make the interpretation of the data more clear. For example, it becomes clear now that pIgR downregulation is not associated with NSP6 (band 6 from the left on the pIgR gel).

Reviewer 2 Report

Comments and Suggestions for Authors

In the manuscript entitled: "SARS-CoV-2 Accessory Protein ORF8 Targets the Dimeric IgA 2 Receptor pIgR" the authors provide a clear and accurate description of the study results on the interaction of SARS-CoV-2 with mucosal immunity.

The study presents a good level of scientific accuracy. It carefully explains the role of ORF8 in downregulating pIgR and its implications for mucosal immunity. The potential mechanisms and the unique characteristics of ORF8 are well-described, and the discussion on the limitations of the study is thorough and informative.

In the study clearly describes the limitations of the study and suggests areas for future research. This adds depth to the discussion and demonstrates a critical understanding of the scope and impact of the study.

Minor revision

The materials and methods section could be improved by adding a graphic workflow that would help the reader have a clearer and smoother view of the different steps.

Line 201 and Line 203: Change FIJI to FIJI-ImageJ

The manuscript is susceptible to publication

Comments on the Quality of English Language

Minor editing of English language are required

Author Response

Reviewer #2

The study presents a good level of scientific accuracy. It carefully explains the role of ORF8 in downregulating pIgR and its implications for mucosal immunity. The potential mechanisms and the unique characteristics of ORF8 are well-described, and the discussion on the limitations of the study is thorough and informative.

In the study clearly describes the limitations of the study and suggests areas for future research. This adds depth to the discussion and demonstrates a critical understanding of the scope and impact of the study.

Minor

The materials and methods section could be improved by adding a graphic workflow that would help the reader have a clearer and smoother view of the different steps.

Line 201 and Line 203: Change FIJI to FIJI-ImageJ

The manuscript is susceptible to publication

Minor editing of English language are required

Response: We would like to thank reviewer #2 for the insightful feedback. We appreciate the positive comments in regard to the depth and detail of the discussion.

To address the comments regarding the Materials and Methods, we have now added an additional supplemental figure detailing the binding and internalization assays (Figure S1). Figure S1 is referenced in line 181-182, “ […] as described in supplemental Figure S1.”, lines 192-195, “Cells incubated with 10 mg/ml IgA-645 and/or 10 mg/ml ORF8-488 diluted in 3% BSA for 30 minutes at 4oC, followed by a subsequent incubation at 37oC for 15 minutes, protected from light, as described in supplemental Figure S1.” lines 371, 395, and 401: “[…] (Figure S1).”

Furthermore, we have changed FIJI to FIJI-ImageJ in lines 217 and 218.

To address the minor editing of the English language, the manuscript was read by a native English speaker and also run through a commercially available text corrector (Antidote V11). Editing of the English language was highlighted in yellow.

Round 2

Reviewer 1 Report

Comments and Suggestions for Authors

I would like to thank the Authors for their response. I appreciate the dedication and the explanations they provide. They have clarified all my concerns. I strongly recommend this manuscript for publication.